# Dynamic Evolution of Aquaculture along the Bohai Sea Coastline and Implications for Eco-Coastal Vegetation Restoration Based on Remote Sensing

**DOI:** 10.3390/plants13020160

**Published:** 2024-01-06

**Authors:** Zhaohua Wang, Kai Liu

**Affiliations:** 1First Institute of Oceanography, MNR, Qingdao 266061, China; wangzhaohua@fio.org.cn; 2Dongying Research Institute for Oceanography Development, Dongying 257000, China; 3Postdoctoral Workstation, National University Science and Technology Park, China University of Petroleum, Dongying 257000, China

**Keywords:** coastal zone, ecosystem restoration, vegetation, NDVI, remote sensing monitoring

## Abstract

The expansion and intensification of coastal aquaculture around the Bohai Sea in China has reduced the tidal flats and damaged the coastal vegetation environment. However, there are few studies on the relationship between the evolution of coastal aquaculture and the variability of coastal vegetation, which limits our understanding of the impact of human activities on the coastal ecosystem. In this study, based on remote sensing technology, we firstly used a combination of a neural network classifier and manual correction to monitor the long-term dynamic changes in aquaculture in the Bohai Sea from 1984 to 2022. We then analyzed its evolution, as well as the relationship between the evolution of coastal aquaculture and the variability of coastal vegetation, in detail. Our study had three main conclusions. Firstly, the aquaculture along the coast of the Bohai Sea showed an expanding trend from 1984 to 2022, with an increase of 538%. Secondly, the spatiotemporal changes in the aquaculture centroids in different provinces and cities varied. The centroid of aquaculture in Liaoning Province was mainly distributed in the Liaodong Peninsula, and moved northwest; that in Hebei Province was distributed in the northeast and moved with no apparent pattern; the centroid of aquaculture in Tianjin was mainly distributed in the southeast and moved westward; and the centroid of aquaculture in Shandong Province was mainly distributed in the northwest and moved in a northwesterly direction. Finally, the expansion of aquaculture of the Bohai Sea has increased the regional NDVI and length of the corresponding coastline, and has made coastlines move toward the sea. Our results provide reliable data support and reference for ecologically managing aquaculture and coastal environmental protection in the Bohai Sea.

## 1. Introduction

Aquaculture is an important component of China’s blue ocean economy, and an important way for humans to obtain high-quality protein and other nutrients. The coastal areas, with their advantages such as low-lying terrain [1,2], abundant freshwater, rich aquatic plants, and seawater resources [3], and ease of management, have become one of the most suitable regions for aquaculture. Expect for breeding, raising, and harvesting fish, shellfish, aquatic plants are also important components of aquaculture around coastal areas. According to the Food and Agriculture Organization of the United Nations (FAO), the production of aquatic plants has reached 35.1 million tonnes accounts, accounting for about 28.6% of global aquaculture [4]. The culture of aquatic plants provides a large amount of food and industrial raw materials for humans. Meanwhile, aquatic plants in aquaculture ponds and areas are also an important food source for aquatic animals [5,6]. In addition, aquaculture plants are also an important buffer in the environment, because they can uptake N and P, thereby alleviating pollution, and supply food and biological conservation sites for fishes and other animals in the coastal environment [7]. Aquatic animals and plant integration in ponds could lead to a sustainable future, lower pollution, and green aquaculture [8]. In addition, aquaculture tends to occupy a large amount of land and mudflats, which can also cause changes in original mudflat vegetation. Especially in coastal areas, where vegetation is relatively fragile, the expansion of coastal aquaculture could lead vegetation habitats to evolve in an unstable direction. Hence, it is important to monitor the evolution of aquatic plants in and around aquaculture ponds and coastal areas.

China’s coastal areas are vast. Since the reform and opening up policy started, increasing demand for food diversity has led to the rapid expansion of aquaculture in China’s coastal regions [9,10]. Among China’s coastal areas, the expansion of aquaculture in the coastal areas of the Bohai Sea is the most typical. The abundant nutrients carried by rivers flowing into the Bohai Sea and the strong consumption capacity of the coastal areas of the Bohai Sea have provided favorable conditions for the rapid expansion of aquaculture over the past 40 years, making the Bohai Sea region one of the most important sources of fishery products in China. However, the rapid expansion of aquaculture ponds has led to a sharp decrease in tidal flats, and to greater water pollution [11,12], bringing great destruction to coastal ecosystems such as the *Suaeda salsa* ecosystem of the Bohai Sea [13]. Degradation of coastal vegetation has led to a decrease in the biodiversity of coastal zones and a gradual increase in seawater pollution in this coastal area [13]. In addition, the aquaculture model in this area is very homogenous and rugged, and does not consider the unity of aquatic plants and aquatic animals [14]. In 2022, the production of aquatic plants in aquaculture around the Bohai Sea was 1.16 million tonnes, accounting for about 42% of China’s total production. However, the distribution of aquatic plant cultures is heterogeneous around the Bohai Sea region: 0.69 million tonnes in Shandong Province, 0.47 million tonnes in Liaoning Province, and no farming in Tianjin City and Hebei Province [15]. Even in Shandong Province, a major aquatic plant culture province, the distribution of aquaculture is still extremely uneven. Most of the aquatic culture is concentrated in the city of Weihai City, Changdao County, and mainly in coastal areas (of water depth > 6 m), and aquatic plant culture in ponds is extremely scarce in this province [16]. Over the past 40 years, the main coastal areas (Water depth > 1 m) have been encroached upon by aquaculture ponds [17]. Degradation has become the dominant evolutionary process in the coastal vegetation system [18]. Meanwhile, the lack of aquatic plant farming in coastal zones with a water depth of less than 1 m has prevented them from fulfilling ecological functions. Therefore, it is necessary to monitor the relationship between the dynamic changes in aquaculture ponds and coastal zone vegetation in the Bohai Sea region over recent decades, to provide a basis for decision making in the management and protection of the coastal ecosystem of the Bohai Sea.

Remote sensing technology has been widely applied to coastal monitoring due to its ability to perform large-scale synchronous observations [19] and low monitoring costs [20,21,22]. For example, it can be used to monitor the coastline [23,24], tidal flats [25,26], and nearshore water bodies [27,28,29]. However, there has been less attention paid to monitoring aquaculture ponds, which have been not the focus of researchers, but rather an indirect target of their studies. For instance, researchers have indirectly monitored aquaculture ponds by studying changes in mangroves or tidal flats [30,31]. In previous studies on monitoring aquaculture ponds, different researchers have focused on different aspects, which can be mainly divided into studies on methods of monitoring aquaculture ponds and studies on dynamic changes in aquaculture ponds [32,33,34,35,36]. In terms of studies on methods of monitoring aquaculture ponds, computer-aided monitoring methods have been widely studied. For example, the U-2-Net deep learning model was used to extract data from aquaculture ponds in the Zhoushan Islands of China [33]; some studies have proposed a framework for extracting aquaculture ponds’ data by integrating existing multi-source remote sensing data from the Google Earth Engine platform, and have used this framework to monitor aquaculture ponds in Shanghai [37]. Studies on dynamic changes in aquaculture ponds have mainly focused on the spatial and temporal distribution of coastal aquaculture ponds over the past few decades, as well as their conversion to other land use types and forms of expansion. For instance, Duan et al. (2021) studied the spatial distribution, changing trends, and driving factors of aquaculture ponds in China’s coastal zone after 1990 [38]. Ren et al. (2019) analyzed the dynamic changes in aquaculture ponds in China’s coastal region from 1984 to 2016, and the impact of other land-use types on the expansion of aquaculture ponds [39]. Luo et al. (2022) analyzed the expansion forms of aquaculture ponds in the coastal areas of Southeast Asia from 1990 to 2015 [35].

Despite the reliable results obtained from the studies, there are still some limitations. On the one hand, as China’s inland sea, the Bohai Sea has seen rapid development of its coastal provinces such as Beijing, Tianjin, Hebei, Shandong, and Liaoning since China’s reform and opening up, and nearshore aquaculture continues to expand. However, existing studies lack timely and long-term monitoring of dynamic changes in aquaculture in the coastal zone of the Bohai sea, as well as a detailed analysis of dynamic changes in aquaculture. Although studies have covered the entire coastal aquaculture of China, their large-scale research cannot simultaneously consider an accurate analysis of dynamic changes in small-scale aquaculture [39,40]. On the other hand, most studies have focused on the driving factors of dynamic changes in aquaculture, and have failed to pay attention to the impacts of aquaculture expansion. Furthermore, studies related to aquatic ponds and coastal zone vegetation change are yet more scarce, limiting the application of this model in analyzing the ecological restoration of coastal zone vegetation. These limitations hinder the protection of the ecological environment and spatial planning of the nearshore area in the Bohai Sea region.

Therefore, the main objectives of this study are (1) to provide a long-term monitoring of aquaculture ponds in the coastal zone of the Bohai sea by using remote sensing technology, and to analyze its spatial–temporal changes in detail; (2) to analyze the impact of aquaculture ponds’ expansion over the past 40 years on the coastline of the Bohai Sea in detail; and (3) to analyze the NDVI of vegetation and the distance of aquatic ponds on the coastline of the Bohai Sea in detail. In our study, we first used a neural network classifier to extract aquaculture ponds from Landsat series images covering the coastal zone of the Bohai Sea from 1984 to 2022, and corrected the extraction results through manual correction to achieve dynamic monitoring of aquaculture ponds in the coastal zone of the Bohai Sea region over the past 40 years. Additionally, we then analyzed the evolution of aquaculture ponds in the coastal zone of the Bohai Sea region and the influence of its expansion on the coastline. Ultimately, we apply this model to the relationship between changes in coastal vegetation and aquatic ponds in the Bohai Sea, which are used to analyze the need for ecological restoration of vegetation in the Bohai Sea coastal zone.

## 2. Study Area

As shown in Figure 1, the Bohai sea is located in the northern part of China, which stretches from 37° N to 41° N and from 117° E to 124° E. It belongs to the temperate monsoon climate zone, with a suitable climate that provides favorable conditions for the growth and reproduction of marine organisms. The Bohai Sea is an inland sea of China, with a sea area of 77,000 km^2^ and an average depth of 18 m. In addition, the influx of the Kuroshio Current (a warm ocean current) branch makes it rich in economic biological resources, and its marine development conditions are superior. The main aquaculture species in the study area include shellfish, shrimp, and crab, as well as trepang. The coastal provinces and cities of the Bohai Sea include Liaoning Province, Hebei Province, Tianjin City, Beijing City, and Shandong Province. Since China’s reform and opening up, the diversity of the coastal economic structure has been enhanced, and the rise of “strengthening from the ocean” has led to the rapid development of the marine economy, as well as the continuous expansion of aquaculture. Although the continuous expansion of aquaculture has brought considerable economic benefits to the coastal areas, it has also effected changes in the Bohai Sea coastline, thereby causing damage to the nearshore ecological environment of the Bohai Sea. Therefore, we chose the Bohai Sea as our study area to investigate dynamic changes in aquaculture and their impact on coastlines from 1984 to 2022. In addition, we conducted field investigations in some areas of the study area in 2020 and 2022, as shown in Figure 1, which shows the location and on-site photos.

## 3. Results

### 3.1. Spatiotemporal Cover Changes in Aquaculture Ponds over the Past 38 Years

In terms of changes in area, as shown in Figure 2, overall, the aquaculture area in the Bohai Sea has shown a year-on-year increasing trend from 1984 to 2022, with the minimum area of 779 km^2^ in 1982, which increased to a maximum area of 4971 km^2^ in 2022, representing an increase of 538%, with an average annual increase of 110 km^2^. We further divided the period from 1984 to 2022 into two time periods: 1984–2012 and 2012–2022, and the trends in area changes in these two time periods were different. From 1984 to 2012, the aquaculture area in the Bohai Sea had increased year by year, from 779 km^2^ to 4555 km^2^ in 2012. However, from 2012 to 2022, the aquaculture area in the Bohai Sea showed a “decrease–increase” trend, decreasing from 2012 to 2017, with an area decrease of 6%, and then increasing from 2017 to 2022. In terms of spatial distribution, as shown in Figure 3, in 1984, aquaculture in the Bohai Sea was mainly distributed in Hebei Province and Tianjin City, with less distribution in Liaoning Province and Shandong Province; in 2022, aquaculture in the Bohai Sea was distributed in all four provinces and cities, mainly in the north and east of Liaoning Province, Hebei Province, Tianjin City, and the western part of Shandong Province. In the past 38 years, aquaculture in the Bohai Sea has expanded both towards the ocean and in the inland direction.

### 3.2. Changes in Aquaculture Area in Different Provinces

We further analyzed the changes in aquaculture area in different provinces and cities along the Bohai Sea coast from 1984 to 2022. As shown in Figure 4, the area changes varied among different provinces and cities. The aquaculture area in the Shandong and Liaoning provinces increased year by year from 146 km^2^ to 1812 km^2^, and from 173 km^2^ to 1447 km^2^, respectively, during the period from 1984 to 2022, increasing by 1146% and 739%, respectively. However, the aquaculture area in Hebei Province and Tianjin City showed an overall trend of “increase–decrease” from 1984 to 2022. The area showed an increasing trend from 1984 to 2007 and 1984 to 2012, respectively, and a decreasing trend from 2007 to 2022 and 2012 to 2022, respectively. The maximum areas for Hebei Province and Tianjin City were reached in 2007 and 2012, respectively, with a maximum area of 1244 km^2^ and 742 km^2^. In terms of proportion, aquaculture in Hebei Province dominated from 1984 to 1987, while Shandong Province took the lead from 1987 to 2022.

### 3.3. Spatiotemporal Changes in Centroids of Aquaculture Ponds over the Past 38 Years

Figure 5 shows the spatiotemporal changes in the centroid of aquaculture ponds in different provinces and cities over the past 38 years, and Table 1 and Table 2 shows the displacement of the centroid of aquaculture ponds in each province and city between 1984 and 2022. From Figure 5 and Table 1 and Table 2, it can be seen that the spatial changes in the centroid of aquaculture ponds in different provinces and cities has differed over the past 38 years. In Liaoning Province, the centroid of aquaculture has mainly been located in the Liaodong Peninsula over the past 38 years, moving overall in the northwest direction, with a maximum displacement of 49 km. This occurred during the period from 2007 to 2012, with the direction of movement being from north to south, indicating that a large number of aquaculture ponds were newly added in the southern part of Liaoning Province from 2007 to 2012. In Hebei Province, the centroid of aquaculture has mainly been located in the northeast part of the province over the past 38 years, and moved near the centroid of aquaculture in 1984, with no obvious direction of movement. The maximum displacement was 28 km, which occurred during the period from 2007 to 2012, while the minimum displacement was 4 km between 1992 and 1997. Over the past 38 years, the centroid of aquaculture ponds in Tianjin has mainly been located in the southeast, moving overall to the west. The maximum displacement occurred during the period from 1992 to 1997, with a distance of 25 km. As for Shandong Province, the centroid of aquaculture has mainly been located in the northwest over the past 38 years, moving overall in the northwest direction. In the early stage, the centroid of the province was mainly located inside the Laizhou Bay, but as time went on, with the appearance of a large number of aquaculture ponds in the northwest of Shandong Province, the centroid also moved in the northwest direction. The maximum displacement occurred during the period from 1992 to 1997, with a distance of 28 km.

### 3.4. The Impact of Aquaculture Expansion on the Bohai Sea Coastline

Table 3 shows the coastline lengths corresponding to aquaculture ponds in each province in 1984 and 2022. According to Table 3, over the past 38 years, the length of coastline corresponding to aquaculture has increased annually, with the total length increasing from 438 km in 1984 to 1838 km in 2022, representing a 320% increase. For each province and city, the length of coastline corresponding to aquaculture increased annually from 1984 to 2022. Among them, the length of coastline corresponding to aquaculture in Liaoning province has increased the most, by 449% over the 38-year period.

We further divided the coastal advancement speed impacted by aquaculture ponds into equally spaced intervals and recorded the length of the coastline within different speed intervals, as shown in Table 4. On one hand, the coastline around the Bohai Sea had advanced towards the sea under the impact of aquaculture, with the coastal advancement speed ranging from 0 m/year to 700 m/year from 1984 to 2022 over 38 years. The highest advancement speed interval of 600 m/year to 700 m/year occurred in Shandong Province, while the advancement speeds in other provinces and cities were between 0 m/year and 300 m/year. On the other hand, the advancement speed of each province was mainly in the range of 0 m/year to 100 m/year, with this speed interval accounting for the largest proportion of the coastal length. The proportions of the advancement speed in the range of 0 m/year to 100 m/year in Liaoning Province, Hebei Province, Tianjin City, and Shandong Province were 71%, 75%, 96%, and 40%, respectively.

### 3.5. The Impact of Coastline Expansion on Average NDVI Values in Coastal Zones

The average NDVI values of in the coastline zone in Laizhou Bay in Shandong Province in 1984 and 2022 are shown in Table 5. Over the past 38 years, the average NDVI values of coastline zone have decreased annually. The total NDVI decreased from 0.48 in 1984 to −0.21 in 2012, representing a 143% decrease. For Shandong Province, the coastline and aquaculture ponds increased annually from 1984 to 2012. However, in the 2022, the NDVI increased with the decrease in the area of aquaculture ponds. This suggests that the disappearance of offshore aquaculture ponds has an important role to play in the restoration of coastal zone vegetation.

In order to study the change in NDVI alongside the area of aquaculture ponds, as shown in Figure 6, we firstly generated buffer zones with intervals of 10 km around an aquaculture pond, and we calculated the average NDVI values within buffer zones of different distances. Overall, as the distance from the aquaculture pond increases, the average value of NDVI also increases. The average NDVI increases from 0.25 within the 10 km buffer zone to 0.83 within the 80 km buffer zone.

## 4. Discussions

As shown in Figure 7, We firstly monitored dynamic changes in aquaculture around the Bohai Sea from 1984 to 2022. Additionally, we then analyzed the spatiotemporal changes in aquaculture ponds, changes in centroids of aquaculture ponds, and impact of aquaculture expansion on coastlines. Finally, based on our results, we proposed two suggestions for coastal management.

In recent decades, especially since 2021, numerous reviews and research articles concerning SA’s roles in plants’ tolerance responses to biotic stresses have been published, and some articles published from 2021 to 2023 and selected from four database sites, including Sciencedirect, Web of Science, Scopus, and MDPI, are listed in Table 1. From Table 1, it can be seen that these selected articles mainly deal with such biotic stress factors as bacteria, fungi, and insects. This study shows that aquaculture in the Bohai Sea has shown an increasing trend over the past 38 years, and centroid shifts in each province and city have varied. The driving forces behind aquaculture in the Bohai Sea can be divided mainly into natural driving forces and social and economic driving forces. On the one hand, in terms of natural driving forces, the main rivers flowing into the Bohai Sea include the Yellow River, Haihe River, Luanhe River, Liaohe River, and Xiaoqing River, which carry a large amount of nutrients into the sea and provide sufficient natural conditions for the expansion of aquaculture in the Bohai Sea. In addition, compared with other coastal industries, the construction cost of aquaculture ponds is relatively low, which further promotes the expansion of aquaculture. On the other hand, in terms of social and economic driving forces, China implemented the policy of reform and opening up in the 1970s, and subsequently, China’s rapid economic development has improved people’s living standards, which has led to an increase in demand for food diversity and promoted the rapid expansion of aquaculture in the Bohai Sea. In fact, according to research on global aquaculture monitoring, the total area of global aquaculture is as high as 55,337 km^2^. Asia, led by China, has the largest distribution area of aquaculture, accounting for 89% of global aquaculture, with China having the highest proportion of 69% [41].

We analyzed the displacement of centroids of aquaculture ponds from 1984 to 2022 in detail. The direction of the displacement of aquaculture ponds’ centroids indicates that aquaculture will increase in that direction. Consequently, environmental factors such as water quality in this area should be given priority in terms of monitoring. After studying the impact of aquaculture’s expansion on the coastline, the results show that the length of coastline corresponding to aquaculture ponds increased year by year, by 320%. Furthermore, aquaculture caused the coastline of the Bohai Sea to advance (Figure 8). Although the expansion of nearshore aquaculture has increased profits, the further advancement of the coastline caused by aquaculture expansion will have a further impact on the environment. On the one hand, it will change the nearshore wave pattern and tidal flow. However, many wetlands have been turned into aquaculture ponds, greatly damaging local environmental quality and resilience to risky disasters.

As the coastline advances, the shape of the coastline will change, thereby altering the direction and energy of nearshore tides and waves. On the other hand, the advancement of the coastline will reduce nearshore tidal flats, causing nearshore sediment to move toward the sea, thus further affecting the nearshore ecosystem [42,43]. The original coastal environment has been altered by aquaculture ponds, leading to the degradation of coastal vegetation. Therefore, scientific planning of the expansion of aquaculture sites along the Bohai Sea coast should be given attention by relevant departments. Furthermore, withdrawing from aquaculture to restore wetlands has led to an increase in shoreline vegetation, suggesting that the protection policy of coastal vegetation based on aquaculture control is much needed.

We found that the lengths of coastlines corresponding to the aquaculture ponds increased from 1984 to 2022, during which time natural coastlines were replaced with artificial coastlines, and further led to a decline in coastline species. However, similar situations are also occurring along tropical coastlines, where unsustainable aquaculture expansion damages the unique mangrove ecosystem along tropical coastlines [41,44]. To ensure long-term sustainability, it is necessary to improve governance frameworks and establish standards for sustainable aquaculture [44]. We therefore propose that emphasis should be placed on the protection of natural coastlines. Additionally, we also propose that a coastal ecological warning model should be established based on meteorological parameters such as wave levels and tidal heights, which will be our future focus.

## 5. Data and Methods

### 5.1. Data

To monitor aquaculture in the study area from 1984 to 2022, we selected Landsat series images from 1984, 1987, 1992, 1997, 2007, 2013, 2017, and 2022, which cover the study area, including Landsat-8, Landsat-7, and Landsat-5 images. A total of 108 images were used, which can be downloaded from http://glovis.usgs.gov (accessed on 12 November 2022). The data identification used is shown in the Appendix A. As most aquaculture ponds are filled with water from May to October, we selected images taken during this period to extract as much aquaculture pond data as possible from the remote sensing images. The spatial resolution of images used is 30 m.

### 5.2. Extracting Aquaculture Ponds’ Cover Data

We used a combination of a neural network classifier and manual correction to extract aquaculture ponds from remote sensing images. Firstly, we used the neural network to coarsely extract aquaculture ponds from the remote sensing images, and then manually corrected the extraction results to improve the final accuracy of the extraction.

In the extraction stage of the neural network classifier, after radiometric calibration and atmospheric correction, we first trained the neural network classifier, using all the bands of the images, the normalized vegetation index (NDVI), and the normalized water index (NDWI) as training features. The formulas for calculating the normalized vegetation index (NDVI) and normalized water index (NDWI) are as follows:(1)$$NDVI=\frac{\rho_{NIR}-\rho_{RED}}{\rho_{NIR}+\rho_{RED}}$$
(2)$$NDWI=\frac{\rho_{GREEN}-\rho_{NIR}}{\rho_{GREEN}+\rho_{NIR}}$$
where *ρ_GREEN_*, *ρ_RED_*, and *ρ_NIR_* represent the images of the green, red, and near-infrared bands, respectively.

After training, Landsat images were inputted into the neural network classifier to obtain the rough extraction results of the aquaculture ponds. As the extraction of aquaculture ponds was essentially the same as the extraction of water bodies, there might be other water bodies present in the rough extraction results obtained by the neural network classifier. Therefore, after rough extraction, we further used manual correction to adjust the rough extraction results of the neural network, and obtain the final accurate result.

### 5.3. Measuring the Dynamic Changes in Aquaculture Ponds

We expressed the expansion of and changes in aquaculture ponds in each province by calculating the centroid of aquaculture ponds in each province over the past 40 years. The formula for calculating the centroid is as follows [45]:(3)$$X=\frac{\sum_{i=1}^{n}A_{i}X_{i}}{\sum_{i=1}^{n}A_{i}}$$
(4)$$Y=\frac{\sum_{i=1}^{n}A_{i}Y_{i}}{\sum_{i=1}^{n}A_{i}}$$
where *X* and *Y* represent the longitude and latitude of the centroid, respectively. n is the total number of patches of aquaculture ponds in each province in a certain year, *A_i_* is the area of the aquaculture pond, and *X_i_* and *Y_i_* are the longitude and latitude of the centroid of the *i*th patch.

### 5.4. Measuring the Impact of Aquaculture Expansion on the Coastline

In this study, we mainly focused on the impact of aquaculture’s expansion on the length of the coastline and the rate of the coastline’s migration.

As for the impact on the length of the coastline, it was measured by calculating the changes in the length of the coastline caused by the changes in aquaculture.

As for the impact on the rate of coastline’s migration, it was measured by calculating the end point rate (EPR) of the coastline corresponding to aquaculture using the Digital Shoreline Analysis System (DSAS 5.1) plugin. The calculation formula is as follows [46]:(5)$$EPR_{ij}=\frac{\Delta d_{ij}}{\Delta t_{ij}}$$
where $EPR_{ij}$ represents the end point rate between the i-th and j-th year; $\Delta d_{ij}$ represents the distance between the profile line and the intersection points of the coastline in the i-th and j-th years; and $\Delta t_{ij}$ represents the time interval between the i-th and j-th years.

### 5.5. Accuracy Assessment

After extracting aquaculture, the overall accuracy, user’s accuracy, producer’s accuracy, and Kappa coefficient were calculated to evaluate the accuracy of the extraction results. In addition, we randomly selected aquaculture in the Yellow River Delta and the coast of Laizhou Bay in the study area, and conducted field investigations at this location in April 2020 and August 2022 to verify the accuracy of the aquaculture extraction. Field points of investigation are shown in Figure 1. The overall accuracy, user’s accuracy, producer’s accuracy, and Kappa coefficient were higher than 0.81, 0.84, 0.87 and 81%, respectively.

## 6. Conclusions

This study used remote sensing technology to monitor the dynamic changes in aquaculture in the Bohai Sea from 1984 to 2022; we analyzed its dynamic evolution over the past 38 years, and its expansion’s impact on the Bohai Sea coastline, in detail. Our conclusions are as follows:(1)The aquaculture area showed a trend of increasing from 1984 to 2022, with an increase of 538%. Spatially, aquaculture was mainly distributed in Hebei Province and Tianjin City, in 1984, but was distributed in Hebei Province, Tianjin City, Liaoning Province, and Shandong Province in 2022(2)The area change trends in different provinces and cities were different. The aquaculture area of Shandong Province and Liaoning Province increased year by year from 1984 to 2022, while the aquaculture area of Hebei Province and Tianjin City showed an overall “increase–decrease” trend from 1984 to 2022.(3)The spatio-temporal changes in aquaculture’s centroids varied among different provinces and cities.(4)The expansion of aquaculture ponds decreased the coastal NDVI, with the pond area increasing from 0 km^2^ to 46 km^2^ and the NDVI decreasing from 0.41 to −0.21. However, the NDVI increased to 0.25 when the coastal ponds decreased to 0 km^2^.(5)The expansion of aquaculture of the Bohai Sea increased the length of the corresponding coastline, which grew by 449% from 1984 to 2022. It also caused the Bohai Sea coastline to advance towards the sea, with most of the coastline advancing at a speed of 0–100 m/a.

In summary, aquaculture extension for the past 40 years has caused great potential risks for eco-coastal zones, where coastal vegetation is reducing and marshy areas with salt aggregation are increasing. Spatiotemporal monitoring for climate, soil, vegetation, key water parameters, and changes in coastal plants (including seabed desertification, phytoplankton, benthic algae, macro-algae, and eutrophication) are necessary for restoring eco-coastal zones and maintaining sustainable productivity.

## Figures and Tables

**Figure 1 plants-13-00160-f001:**
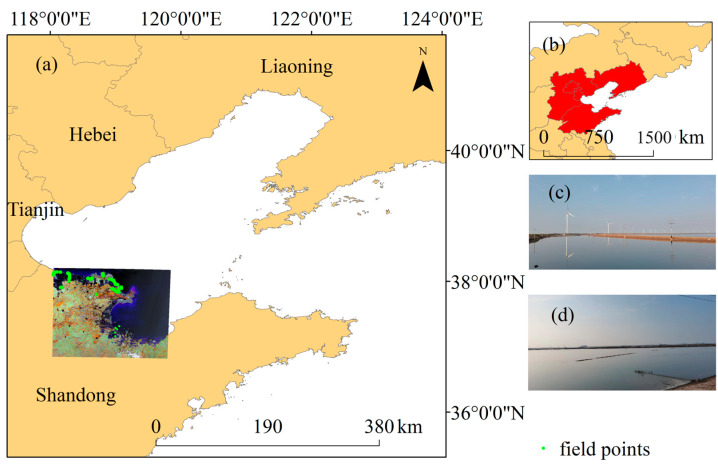
Study area. (**a**) enlarged view of study area; (**b**) location of study area; (**c**,**d**) photos of aquaculture ponds.

**Figure 2 plants-13-00160-f002:**
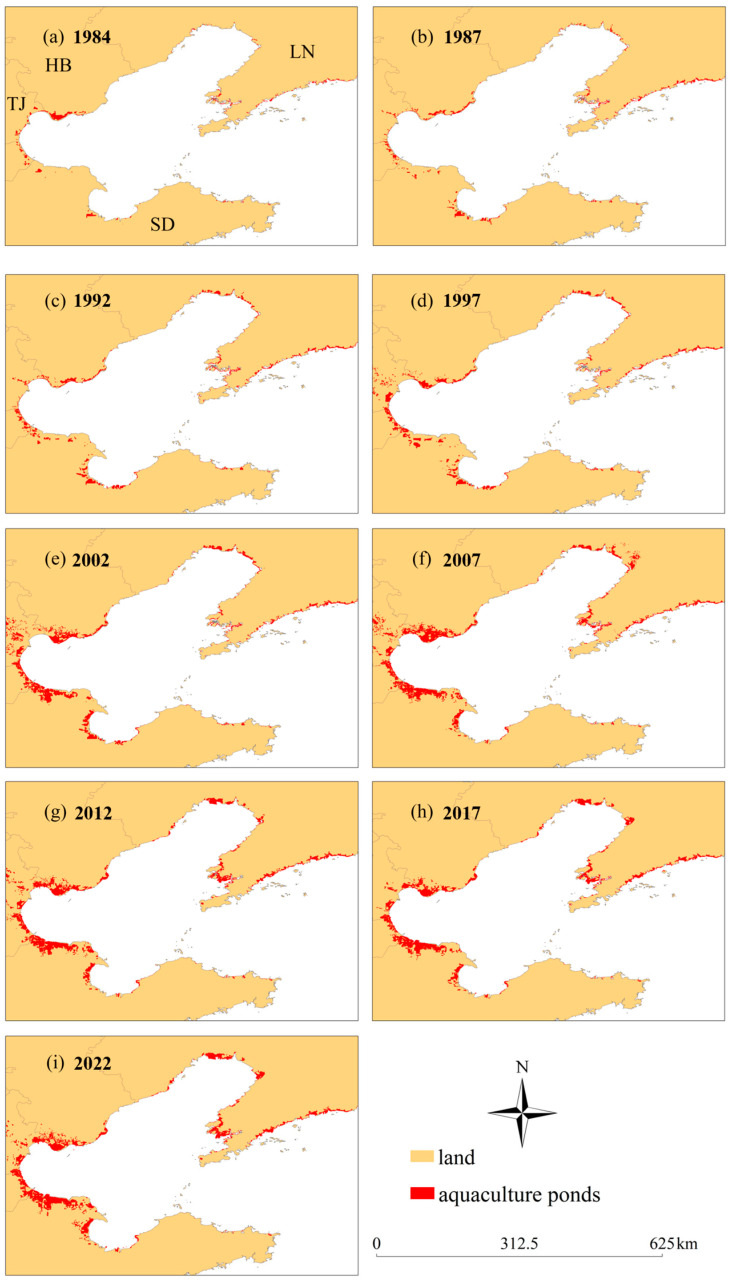
Spatial distribution of aquaculture in the Bohai Sea Area from 1984 to 2022. (**a**–**i**) Spatial distributions of aquaculture ponds in 1984, 1987, 1992, 1997, 2002, 2007, 2012, 2017, and 2022, respectively. HB, represent Hebei Province; LN, represent Liaoning Province; TJ represent Tianjin City; SD represent Shandong Province.

**Figure 3 plants-13-00160-f003:**
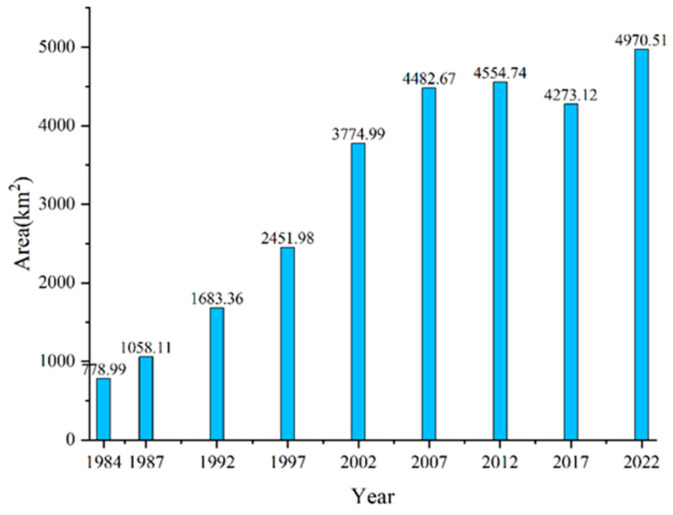
Aquaculture area in the Bohai Sea Area from 1984 to 2022.

**Figure 4 plants-13-00160-f004:**
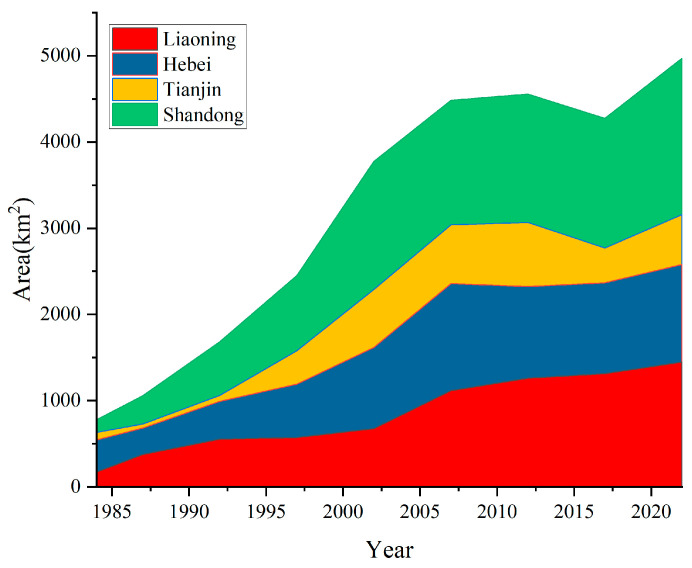
Spatial distribution of aquaculture in the Bohai Sea Area from 1984 to 2022.

**Figure 5 plants-13-00160-f005:**
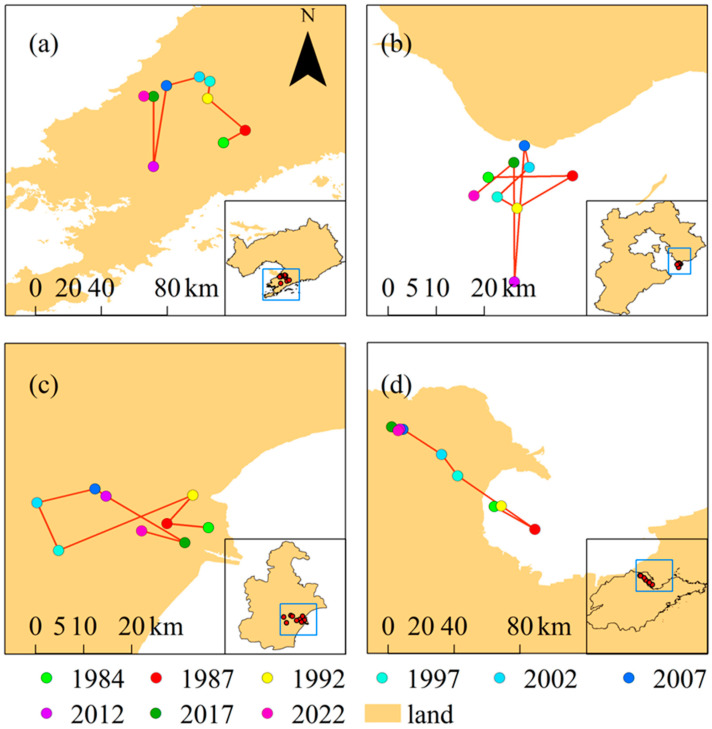
Spatiotemporal changes in the centroid of aquaculture ponds in different provinces and cities over the past 38 years. (**a**–**d**) Spatiotemporal changes in the centroid of aquaculture ponds in Liaoning Province, Hebei Province, Tianjin City, and Shandong Province, respectively.

**Figure 6 plants-13-00160-f006:**
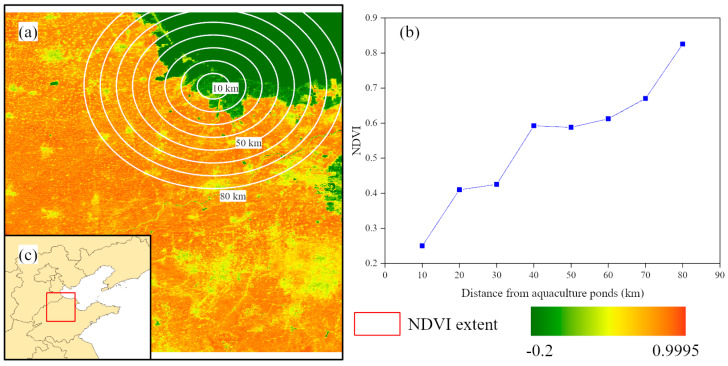
(**a**) NDVI image. (**b**) Average NDVI values in different distance buffers. (**c**) Location of NDVI image.

**Figure 7 plants-13-00160-f007:**
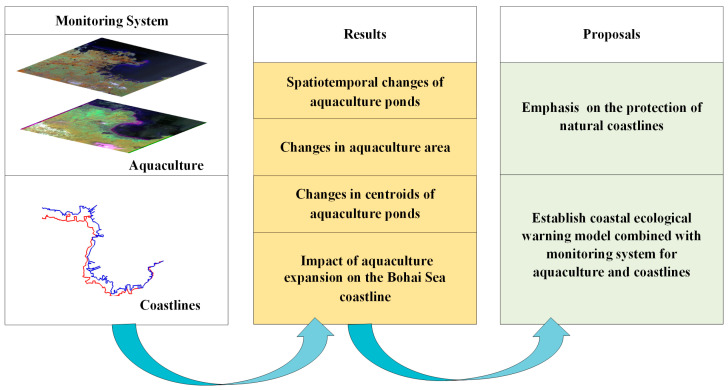
Framework of this study.

**Figure 8 plants-13-00160-f008:**
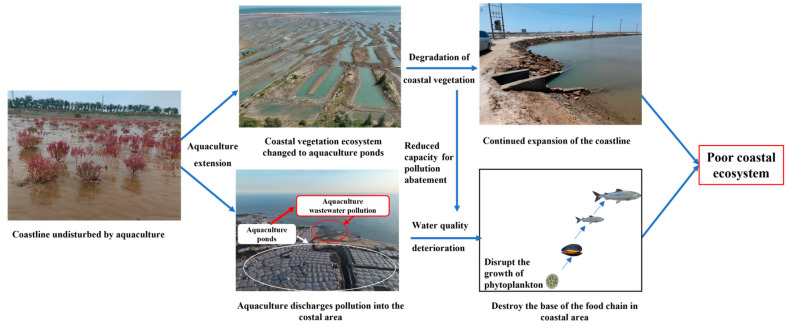
Effect of increased aquaculture on the coastline and coastal ecosystem.

**Table 1 plants-13-00160-t001:** The displacement distance (in kilometers) of the centroid of aquaculture ponds in each province between 1984 and 2002.

	1984–1987	1987–1992	1992–1997	1997–2002
Liaoning	13 km	26 km	10 km	6 km
Hebei	14 km	11 km	4 km	8 km
Tianjin	7 km	7 km	25 km	10 km
Shandong	24 km	22 km	28 km	15 km

**Table 2 plants-13-00160-t002:** The displacement distance (in kilometers) of the centroid of aquaculture ponds in each province between 2002 and 2022.

	2002–2007	2007–2012	2012–2017	2017–2022
Liaoning	16 km	49 km	42 km	4 km
Hebei	5 km	28 km	25 km	9 km
Tianjin	10 km	2 km	16 km	7 km
Shandong	24 km	1 km	4 km	4 km

**Table 3 plants-13-00160-t003:** The total coastline lengths (in kilometers) occupied by the aquaculture ponds for each province and city in 1984 and 2022.

	1984	2022
Liaoning	219 km	1205 km
Hebei	104 km	226 km
Tianjin	36 km	51 km
Shandong	79 km	356 km

**Table 4 plants-13-00160-t004:** The total coastline lengths (in kilometers) occupied by the aquaculture ponds for each province and city within different speed intervals (in meters per year).

	0–100 (m/y)	100–200 (m/y)	200–300 (m/y)	300–400 (m/y)	400–500 (m/y)	500–600 (m/y)	600–700 (m/y)
Liaoning	322 km	76 km	58 km	0 km	0 km	0 km	0 km
Hebei	139 km	32 km	0 km	13 km	0 km	0 km	0 km
Tianjin	44 km	2 km	0 km	0 km	0 km	0 km	0 km
Shandong	102 km	37 km	69 km	32 km	0 km	0 km	15 km

**Table 5 plants-13-00160-t005:** The coastline length and aquaculture pond area corresponding to the NDVI of the coastline and for Shandong Province in 1984 and 2022.

	1984	2012	2022
NDVI	0.48	−0.21	0.25
Coastline	78 km	283 km	356 km
Area	0 km^2^	45 km^2^	0 km^2^

## Data Availability

Data are contained within the article and Appendix A.

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
