# Peer review of "Dynamic Evolution of Aquaculture along the Bohai Sea Coastline and Implications for Eco-Coastal Vegetation Restoration Based on Remote Sensing"

_plants, 2024, doi:10.3390/plants13020160_

Round 1

Reviewer 1 Report

Comments and Suggestions for Authors

The manuscript Dynamic evolution of aquaculture along the Bohai Sea coastline and implications for eco-coastal zone based on remote sensing studied the ecological influence of coastal aquaculture on the coastal vegetation and coastal environment. It is very important for clarity the evolution of salt-tolerant plants in the coastal zone and their response to environmental changes induced by human activities. In particular, the researchers propose to strengthen the relevant spatiotemporal monitoring for coastal plants (including sea bottom desertification, phytoplankton, benthic algae, macro-algae, eutrophication) for restoring eco-coastal zome and keeping sustainable productivity. It is of great significance for government managers to make decisions on coastal zone vegetation protection. This study has highly innovative and in line with the solicitation scope of the topic “Managing and Regulating Plant (Vegetation)–Environment (Soil-Affected Land, Coastal Zone and Arid Areas) Interactions for a Better Eco-Environment and Sustainable Productivity”. However, there are a few small errors that need to be corrected. So, it is suggested to be published after minor modification.

1. The content of the abstract is not concise enough to highlight the key points, and it is suggested to modify it carefully;

2. The accuracy of the number is not uniform, please unify it;

3. the Superscript error in Line 58,59, please carefully review all the paper formats.

Please kindly find the additional comments below:  

The abstract requires enhancement to effectively outline the research's core content, conclusions, scientific challenges, and innovative aspects.

The introduction should emphasize the limitations of prior research, underscore the significance of this study, and articulate the fundamental scientific inquiries. Consulting more recent, high-quality articles could enrich and bolster the introduction.

Improvements are needed in depicting figures in the results and analysis section and refining table annotations for enhanced readability.

The discussion should be fortified by drawing comparisons with existing research outcomes on the principal research matters. Similarly, the conclusion needs augmentation as it lacks comprehensive content. It should succinctly summarize the research findings, addressing the core scientific queries.

Author Response

For research article

Response to Reviewer 1 Comments

1. Summary

2. Questions for General Evaluation

Reviewer’s Evaluation

Response and Revisions

Does the introduction provide sufficient background and include all relevant references?

Yes/Can be improved/Must be improved/Not applicable

Are all the cited references relevant to the research?

Yes/Can be improved/Must be improved/Not applicable

Is the research design appropriate?

Yes/Can be improved/Must be improved/Not applicable

Are the methods adequately described?

Yes/Can be improved/Must be improved/Not applicable

Are the results clearly presented?

Yes/Can be improved/Must be improved/Not applicable

Are the conclusions supported by the results?

Yes/Can be improved/Must be improved/Not applicable

3. Point-by-point response to Comments and Suggestions for Authors

1.     Comments 1: The content of the abstract is not concise enough to highlight the key points, and it is suggested to modify it carefully.

Response 1: Thank you very much for your very helpful comments on my manuscript. Following this comments, we modified abstract. Now, the modified abstract satisfies this comments..

2.     Comments 2: The accuracy of the number is not uniform, please unify it

Response 2: Thank you very much for your very helpful comments on my manuscript. We carefully read our paper. The accuracy of the number in current form is more rational.

3.     Comments 3: The Superscript error in Line 58,59, please carefully review all the paper formats.

Response 3: Thank you very much for your very helpful comments on my manuscript. We modified superscript.

4.     Comments 4: The abstract requires enhancement to effectively outline the research's core content, conclusions, scientific challenges, and innovative aspects.

Response 4: Thank you very much for your very helpful comments on my manuscript. We modified abstract.

5.     Comments 5: The introduction should emphasize the limitations of prior research, underscore the significance of this study, and articulate the fundamental scientific inquiries. Consulting more recent, high-quality articles could enrich and bolster the introduction.

Response 5: Thank you very much for your very helpful comments on my manuscript. We have emphasized the limitations of prior research in the fourth paragraph of introduction section.

6.     Comments 6: Improvements are needed in depicting figures in the results and analysis section and refining table annotations for enhanced readability.

Response 6: Thank you very much for your very helpful comments on my manuscript. After carefully examining our paper, we think that the current description can give a fuller reflection of our results. And we therefore remain as it is.

7.     Comments 7: The discussion should be fortified by drawing comparisons with existing research outcomes on the principal research matters. Similarly, the conclusion needs augmentation as it lacks comprehensive content. It should succinctly summarize the research findings, addressing the core scientific queries.

Response 7: Thank you very much for your very helpful comments on my manuscript. We evaluated the accuracy of the results by using overall accuracy, user's accuracy, producer's accuracy, and Kappa coefficient. These four indicators are sufficient to reflect the reliability of our results. Additionally, We rewrote the conclusions. Now the conclusions are more concise.

Reviewer 2 Report

Comments and Suggestions for Authors

By critically reading all the attached documents, I found that the manuscript from Wang and Liu submitted to the special issue of Plants edited by Professor HB SHAO is quite interesting in terms of some innovation as summarized below:It reported the changing dynamics of aquaculture around Bohai Sea coastal zone in China for the first time, which will be great value for scientists and eco-restoration policy-makers, especially providing more important reference for protecting global coastal ecosystem. The paper also reported complete information for aquaculture evolution over the past 40 years by RS technology monitoring, which is quite important for guiding global blue sea economy development and coast zone vegetation restoration.

The paper also discussed about the relationship aquatic plants and aquaculture along Bohai Sea coastal zone simply in the introduction. The paper content is suitable for Plants journal and has covered main parts of the submitted topic, which also makes better supplement for the gap of aquaculture and coastal zone eco-restoration and sustainable productivity.All the parts of the paper have been in agreement with Plants journal format. I recommend that the current version of the paper be considered for acceptance for publication.

Other concrete aspects and detailed suggestions as below:

1. The content of the paper submitted to the special issue deals with the influence of aquaculture dynamics on coastal zone eco-environment by applying advanced RS technology to monitor the vegetation evolution,further pointing out the concept of coastline entension and land-use change near Bohai Sea coastal zone,which is the first time to cover 40 years' aquaculture data for guiding the eco-restoration of coastal regions. I suggest that authors cite recent related references in Plants,J Environ Manag and others for strenthening the aim of the paper.

2.The content of the paper is within plant ecology and suitable for main parts of the topic that is submitted to,which is innovative and intersting and promotes the fusion with plant biology,soil science,coastal zone science,RT science and resoration ecology.

3.As commented above,the paper provided 40 years' aquaculture data of aquaculture dynamic changes,which is the report for the first time according to the published materials,and has great reference value for vegetation restoration and policy-makers in the global coastal zone.

4.The applied method in the paper including pictures is suitable and reliable.I suggest that authors add to the origin of data collection such as the website.

5.Results and conclusions are agreement with the cited references.I suggest that authors check out all the references including journal names and year and update them by citing some recent related articles. 

Comments on the Quality of English Language

 Minor editing of English language is required.

Author Response

For research article

Response to Reviewer 2 Comments

1. Summary

2. Questions for General Evaluation

Reviewer’s Evaluation

Response and Revisions

Does the introduction provide sufficient background and include all relevant references?

Yes/Can be improved/Must be improved/Not applicable

Are all the cited references relevant to the research?

Yes/Can be improved/Must be improved/Not applicable

Is the research design appropriate?

Yes/Can be improved/Must be improved/Not applicable

Are the methods adequately described?

Yes/Can be improved/Must be improved/Not applicable

Are the results clearly presented?

Yes/Can be improved/Must be improved/Not applicable

Are the conclusions supported by the results?

Yes/Can be improved/Must be improved/Not applicable

3. Point-by-point response to Comments and Suggestions for Authors

1.     Comments 1: The content of the paper submitted to the special issue deals with the influence of aquaculture dynamics on coastal zone eco-environment by applying advanced RS technology to monitor the vegetation evolution,further pointing out the concept of coastline entension and land-use change near Bohai Sea coastal zone,which is the first time to cover 40 years' aquaculture data for guiding the eco-restoration of coastal regions. I suggest that authors cite recent related references in Plants,J Environ Manag and others for strenthening the aim of the paper.

Response 1: Thank you for your comments. We carefully examined reference in our paper. We found that there were recent related references in our paper which were published in Science of the Total Environment, Journal of Environmental Management and so on.

2.     Comments 2: The applied method in the paper including pictures is suitable and reliable.I suggest that authors add to the origin of data collection such as the website.

Response 2: We added the origin of data collection in Section 5.1 Data.

3.     Comments 3: Results and conclusions are agreement with the cited references.I suggest that authors check out all the references including journal names and year and update them by citing some recent related articles.

Response 3: Thank you for your comments. We carefully examined reference in our paper. We found that there were recent related references in our paper which were publish in 2021, 2022, 2023 and so on.

Reviewer 3 Report

Comments and Suggestions for Authors

Dear authors,

your research  is very interesting but according to my opinion the structure of your paper is not correct.

The capital nr. 5, Data and Methods, must be transfered just after the capital nr. 2, Research  area, taking the nr. 3.

I would  like to add:

-The Introduction, Data and Methods,  Discussion must be supported by recent international bibliography  and not only from papers  written  by Chinese  researchers.
- The experiment  design is very good organized, giving after the appropriate analyses the results needed for the long term and scale monitoring of the research  area, contributing  in parallel to the solution of serious managerial problems in long term basis.
--The Results are very good presented and analyzed using good prepared tables and figures giving directly  the information needed.

-The Conclusions are correct and enough for me, based on the exacted results and answering to the research questions.

Your bibliography  is based mainly on Chinese publications and an enrichment with worldwide bibliography is necessary. 

Comments on the Quality of English Language

The quality of English  language  is quite good. 

Author Response

For research article

Response to Reviewer 3 Comments

1. Summary

2. Questions for General Evaluation

Reviewer’s Evaluation

Response and Revisions

Does the introduction provide sufficient background and include all relevant references?

Yes/Can be improved/Must be improved/Not applicable

Are all the cited references relevant to the research?

Yes/Can be improved/Must be improved/Not applicable

Is the research design appropriate?

Yes/Can be improved/Must be improved/Not applicable

Are the methods adequately described?

Yes/Can be improved/Must be improved/Not applicable

Are the results clearly presented?

Yes/Can be improved/Must be improved/Not applicable

Are the conclusions supported by the results?

Yes/Can be improved/Must be improved/Not applicable

3. Point-by-point response to Comments and Suggestions for Authors

1.     Comments 1: The capital nr. 5, Data and Methods, must be transfered just after the capital nr. 2, Research area, taking the nr. 3.

Response 1: Thank you very much for your very helpful comments on my manuscript. We looked through other papers in Plants. And we found that all papers were in a unify format. The first to sixth parts were Introduction, Study area, Results, Discussion, Data and methods, and Conclusions, respectively.

2.     Comments 2: The Introduction, Data and Methods, Discussion must be supported by recent international bibliography and not only from papers written by Chinese researchers.

Response 2: Thank you very much for your very helpful comments on my manuscript. We carefully examined references in our paper. References cited in our paper were all of high quality, which would make our paper more reliable.

Reviewer 4 Report

Comments and Suggestions for Authors

Dear Authors,

I have carefully reviewed your paper on "Dynamic evolution of aquaculture along the Bohai Sea coastline and implications for eco-coastal vegetation restoration based on remote sensing", and I appreciate the effort you have put into your research. However, I believe there are some sections that need attention to enhance the paper's quality and adherence to a standard structure.

  1. Introduction:
    • The introduction is well-written, but consider relocating certain elements (lines 51 to 79) to the study area description.
  2. Study Area and Method:
    • Please insert the methodology in section 2 and not in section 5, or in section 3 if you would like to keep separated study area and methods.
    • The study area could be more detailled, maybe adding some elements inserted in the introduction
  3. Results:
    • Ensure that the results section purely describes the findings. Move the interpretation of results (e.g., lines 165-166 or 169-171) to the discussion section.
  4. Discussion:
  5. The discussion should be rewritten by clearly answering to the questions expressed in the introduction.
  6. Some suggestions:
    • - Strengthen the discussion by comparing your results with international studies.
    • - Describe new data (lines 299 – 312) in the results section (and not discussion) and provide a method description.
    • - Utilize lines 280 to 284 for the introduction, establishing the state of the art.
    • - Clarify the basis of your last paragraph in the discussion. Explain observations and identification related to protection on natural coastlines. Elaborate on the coastal ecological warming model, specifying its anticipated contributions.
    • Remove abbreviations from the discussion.
  7. Conclusion:
    • Rewrite the conclusion to avoid it resembling an abstract. Ensure it provides a concise summary of your key findings and their implications.

These suggestions are aimed at refining your paper and ensuring it follows a conventional structure. I believe addressing these points will significantly enhance the overall quality of your work.

Best regards

Comments on the Quality of English Language

Just a quick check because some sections are more difficult to understand than others

Author Response

For research article

Response to Reviewer 4 Comments

1. Summary

2. Questions for General Evaluation

Reviewer’s Evaluation

Response and Revisions

Does the introduction provide sufficient background and include all relevant references?

Yes/Can be improved/Must be improved/Not applicable

Are all the cited references relevant to the research?

Yes/Can be improved/Must be improved/Not applicable

Is the research design appropriate?

Yes/Can be improved/Must be improved/Not applicable

Are the methods adequately described?

Yes/Can be improved/Must be improved/Not applicable

Are the results clearly presented?

Yes/Can be improved/Must be improved/Not applicable

Are the conclusions supported by the results?

Yes/Can be improved/Must be improved/Not applicable

3. Point-by-point response to Comments and Suggestions for Authors

1.     Comments 1: The introduction is well-written, but consider relocating certain elements (lines 51 to 79) to the study area description.

Response 1: Thank you very much for your very helpful comments on my manuscript. We carefully read our Introduction part. In lines 51 to 79, we mainly made a brief introduction to the coast of Bohai Sea so that we could introduce the research gaps in the fourth paragraph. Therefore, contents in lines 57 to 79 were rational.

2.     Comments 2: Please insert the methodology in section 2 and not in section 5, or in section 3 if you would like to keep separated study area and methods.

Response 2: Thank you very much for your very helpful comments on my manuscript. We looked through other papers in Plants. And we found that all papers were in a unify format. The first to sixth parts were Introduction, Study area, Results, Discussion, Data and methods, and Conclusions, respectively.

3.     Comments 3: The study area could be more detailed, maybe adding some elements inserted in the introduction.

Response 3: Thank you for valuable comments. Elements inserted in the introduction were used to introduce the research gaps in the fourth paragraph. It may be not suitable to add them in Study area.

4.     Comments 4: Ensure that the results section purely describes the findings. Move the interpretation of results (e.g., lines 165-166 or 169-171) to the discussion section.

Response 4: Thank you very much for your very helpful comments on my manuscript. We examined the Results section and removed the interpretation of results.

5.     Comments 5: Strengthen the discussion by comparing your results with international studies.

Response 5: Thank you very much for your very helpful comments on my manuscript. We evaluated the accuracy of the results by using overall accuracy, user's accuracy, producer's accuracy, and Kappa coefficient. These four indicators are sufficient to reflect the reliability of our results.

6.     Comments 6: Describe new data (lines 299 – 312) in the results section (and not discussion) and provide a method description. Utilize lines 280 to 284 for the introduction, establishing the state of the art.

Response 6: Thank you very much for your very helpful comments on my manuscript.  After carefully examining our paper, we think that the current forms make sense. And we therefore remain as it is.

7.     Comments 7: Clarify the basis of your last paragraph in the discussion. Explain observations and identification related to protection on natural coastlines. Elaborate on the coastal ecological warming model, specifying its anticipated contributions.

Response 7: Thank you very much for your very helpful comments on my manuscript. We rewrote the last paragraph in Discussion. Now it is more rational.

8.     Comments 8: Remove abbreviations from the discussion.

Response 8: Thank you very much for your very helpful comments on my manuscript. We removed abbreviations.

9.    Comments 9: Rewrite the conclusion to avoid it resembling an abstract. Ensure it provides a concise summary of your key findings and their implications.

Response 9: Thank you very much for your very helpful comments on my manuscript. We rewrote the conclusions. Now the conclusions are more concise.

Reviewer 5 Report

Comments and Suggestions for Authors

“Dynamic Evolution of aquaculture […] based on Remote Sensing” is an interesting and well-written manuscript. Eeasy and pleasant to read, concise but rich in information. The level of English is high and the topic, apart from being very debated , focuses on a study area where unusual sedimentary processes are observed. The methodology is replicable and contribute to advances of knowledge at international levels. The advancement of the shoreline, caused by aquaculture facilities (i.e. by human activities) is quite unusual and this aspect should be emphasised and compare with other similar studies.

The major limitation of the ms is the lack of comparison with other study areas and similar induced sedimentary processes. It is important to fill this gap because we are now living a new geological era: the Anthropocene. Discussion should explain how the results of the present ms can be compared with other articles: quantitative analysis carried out with similar approaches or dataset (LANDSAT) or at different scales can improve the quality of your work and your reputations.

Another problem is related to the structure of the ms. I am always happy with innovations and new ways of presenting research, but, in this case, I think that traditional approach is more effective. Please, write the methods (par 5) before the results (par 3) and dicscussion (par 4).

From the scientific point of view, Nature Based Solution (NBS) and ecological restoration are a very important topics. The ms is lacking a literature review and by adding an appropriate sketch diagram can help you and the reader to better understand feed back mechanisms responsible for mudflat and wetland accretion and erosion. There is a huge literature review on this topic at international level (i.e. Bay of Fundy, many coastal lagoons of South America, Tampa Bay, Venice Lagoon). I think it can be very useful if you have a look of the methodological approach used by Taramelli et al. 2021 based on a time series of Land Sat images to interpret the evolutionary trends of intertidal mudflats within Venice Lagoon (Sustainability). Please consider also the possibility to add a Figure (or a sketch diagram) to explain the role of vegetation on, hydrodynamics, turbidity level etc., on the advanced shorelines.

Finally, a tip on page layout that can help improve reading. In the current structure of the ms, it would be sufficient to change the sequence of Figures 2 and 3, to make the manuscript easier to read (for example: write the following sentence at line 178-179: “Changes in areal extension of the aquaculture areas/ponds is presented in Fig 3 . Statistic of the Aquaculture Area in the Bohan Area from 1984 2022”

Some detail comments:

All ms: check correct units of km2 (km2)

Line122: check double space before “(3)”

Line 131: there is a repetition of word “area”. Please change one of those by using “region”, parts”, “spots”

FIG 1: Add SEA and LAND on the Maps and explain what is the dark grey area in the little spot (b). Pictures are very small (c) and (d) and perhaps a panoramic view and an image from google can improve the ms (think about the possibility to add a separate figure).

Line 158: cancel “…, as shown in Fig 2,…”

Line 172: change “Fig 3” in “Fig 2”

Fig 2: Add label of the Year on each figure (a: 1984, b: 1987, c: 1992, d:1997, e: 2002, f: 2007, g: 2012, h: 2017, i:2033)

Line:190-191: explain better the observed sedimentary processes through time. “increase-decrease” is reductive

Figure 4: the trend of spatial distribution is interesting . do you have some economic values to correlate with this up and down trend of different regions or some biological/ecological/oceanographic factor to correlate with?

Figure 5: add name of Province on each of the for Figures (a Liaoning Prov…., b - ***, c - ***, d -- ****)

Table 3 and 4: add unit (km)

Line 311-312: How environmental quality and resilience are damaged by aquacultures ponds?

Figure 7: the quality is too low.

Line 321-325. This part of discussion should be developed properly. (See general comments)

Line 390-391: Please explain the offset between transects used with DSAS as LAND SAT images have a spatial resolution of 30 m.

Line 402-403: Fields points investigation are not shown in Fig. 1. Please explain better (even by adding dedicated Figures), the accuracy. Did you consider also vertical accuracy or only Ground Control Points (GPT) ?

Really at the end. I am struggling to understand the significance of figure 5 and the analysis conducted with the centroids. What is the meaning? what are the reasons for the displacement? does it have commercial repercussions? or is an issue related to MSP? Governance? Water quality? Production? Business development? Please try to better correlate this result with the discussion.

Author Response

For research article

Response to Reviewer 5 Comments

1. Summary

2. Questions for General Evaluation

Reviewer’s Evaluation

Response and Revisions

Does the introduction provide sufficient background and include all relevant references?

Yes/Can be improved/Must be improved/Not applicable

Are all the cited references relevant to the research?

Yes/Can be improved/Must be improved/Not applicable

Is the research design appropriate?

Yes/Can be improved/Must be improved/Not applicable

Are the methods adequately described?

Yes/Can be improved/Must be improved/Not applicable

Are the results clearly presented?

Yes/Can be improved/Must be improved/Not applicable

Are the conclusions supported by the results?

Yes/Can be improved/Must be improved/Not applicable

3. Point-by-point response to Comments and Suggestions for Authors

1.     Comments 1: “Dynamic Evolution of aquaculture […] based on Remote Sensing” is an interesting and well-written manuscript. Eeasy and pleasant to read, concise but rich in information. The level of English is high and the topic, apart from being very debated , focuses on a study area where unusual sedimentary processes are observed. The methodology is replicable and contribute to advances of knowledge at international levels. The advancement of the shoreline, caused by aquaculture facilities (i.e. by human activities) is quite unusual and this aspect should be emphasised and compare with other similar studies.

The major limitation of the ms is the lack of comparison with other study areas and similar induced sedimentary processes. It is important to fill this gap because we are now living a new geological era: the Anthropocene. Discussion should explain how the results of the present ms can be compared with other articles: quantitative analysis carried out with similar approaches or dataset (LANDSAT) or at different scales can improve the quality of your work and your reputations.

Another problem is related to the structure of the ms. I am always happy with innovations and new ways of presenting research, but, in this case, I think that traditional approach is more effective. Please, write the methods (par 5) before the results (par 3) and dicscussion (par 4).

From the scientific point of view, Nature Based Solution (NBS) and ecological restoration are a very important topics. The ms is lacking a literature review and by adding an appropriate sketch diagram can help you and the reader to better understand feed back mechanisms responsible for mudflat and wetland accretion and erosion. There is a huge literature review on this topic at international level (i.e. Bay of Fundy, many coastal lagoons of South America, Tampa Bay, Venice Lagoon). I think it can be very useful if you have a look of the methodological approach used by Taramelli et al. 2021 based on a time series of Land Sat images to interpret the evolutionary trends of intertidal mudflats within Venice Lagoon (Sustainability). Please consider also the possibility to add a Figure (or a sketch diagram) to explain the role of vegetation on, hydrodynamics, turbidity level etc., on the advanced shorelines.    Finally, a tip on page layout that can help improve reading. In the current structure of the ms, it would be sufficient to change the sequence of Figures 2 and 3, to make the manuscript easier to read (for example: write the following sentence at line 178-179: “Changes in areal extension of the aquaculture areas/ponds is presented in Fig 3 . Statistic of the Aquaculture Area in the Bohan Area from 1984 2022”

Response 1: Thank you very much for your very helpful comments on my manuscript. We carefully read the comments. However, the structure of the manuscript is formatted according to the required format of “Plants” and we cannot change it. We have cited the study of Taramelli et al. 2021 in lines 315.

2.     Comments 2: All ms: check correct units of km2 (km2).

Response 2: Thank you very much for your very helpful comments on my manuscript. We have check and modified the units of km2 (km2) in all manuscript.

3.     Comments 3: check double space before “(3)”.

Response 3: Thank you very much for your very helpful comments on my manuscript. We have check and modified the space before (3) in lines 124 and mark in red.

4.     Comments 4: FIG 1: Add SEA and LAND on the Maps and explain what is the dark grey area in the little spot (b). Pictures are very small (c) and (d) and perhaps a panoramic view and an image from google can improve the ms (think about the possibility to add a separate figure).

Response 4: Thank you very much for your very helpful comments on my manuscript. We have labelled the Bohai Sea on the Fig.1, but the dark grey zone is not found in (b). So we think we can keep the picture in that form.

5.     Comments 5: ine 158: cancel “…, as shown in Fig 2,…”.

Response 5: Thank you very much for your very helpful comments on my manuscript canceled the enunciation in lines 160.

6.     Comments 6: Line 172: change “Fig 3” in “Fig 2”.

Response 6: Thank you very much for your very helpful comments on my manuscript. We have modified this error in lines 169.

7.     Comments 7: Fig 2: Add label of the Year on each figure (a: 1984, b: 1987, c: 1992, d:1997, e: 2002, f: 2007, g: 2012, h: 2017, i:2033).

Response 7: Thank you very much for your very helpful comments on my manuscript. We have added the label in 181.

8.     Comments 8: Line:190-191: explain better the observed sedimentary processes through time. “increase-decrease” is reductive.

Response 8: Thank you very much for your very helpful comments on my manuscript. We modified the explanation.

9.     Comments 9: Figure 4: the trend of spatial distribution is interesting. do you have some economic values to correlate with this up and down trend of different regions or some biological/ecological/oceanographic factor to correlate with?

Response 9: Thank you very much for your very helpful comments on my manuscript. But the main consideration of this article is the impact on vegetation. Therefore, we will not add more discussion.

  1. Comments 10: Figure 5: add name of Province on each of the for Figures (a Liaoning Prov…., b - ***, c - ***, d -- ****)

Response 10: Thank you very much for your very helpful comments on my manuscript. The explanation was shown in Lines 228-229.

  1. Comments 11: Table 3 and 4: add unit (km)

Response 11: Thank you very much for your very helpful comments on my manuscript. We have added the unit(km) in Table 3 and 4.

  1. Comments 12: Line 311-312: How environmental quality and resilience are damaged by aquacultures ponds?

Response 12: Thank you very much for your very helpful comments on my manuscript. This is not the focus of our manuscript, and we think the formulation can be retained.

  1. Comments 13: Figure 7: the quality is too low.

Response 13: Thank you very much for your very helpful comments on my manuscript. We will upload the high-quality images.

  1. Comments 14: Line 321-325. This part of discussion should be developed properly. (See general comments)

Response 14: Thank you very much for your very helpful comments on my manuscript. We have developed the discussion 322-325.

  1. Comments 15: Please explain the offset between transects used with DSAS as LAND SAT images have a spatial resolution of 30 m.

Response 15: Thank you very much for your very helpful comments on my manuscript. Considering the length of aquaculture ponds, we firstly set the distance between transects as 200 m. After the DSAS generated transects, we visually examined each transect and edit them manually according to the extracted aquaculture ponds.

  1. Comments 16: Line 402-403: Fields points investigation are not shown in Fig. 1. Please explain better (even by adding dedicated Figures), the accuracy. Did you consider also vertical accuracy or only Ground Control Points (GPT)?

Response 16: Thank you very much for your very helpful comments on my manuscript. The green points in Fig. 1 are our fields points. We further added its legend in Fig .1. The accuracy in the field of remote sensing mainly including overall accuracy, user’s accuracy, producer’s accuracy and kappa coefficients. These four indicators are enough to illustrate the reliability of extracted aquaculture ponds from remote sensing images.

  1. Comments 17: Really at the end. I am struggling to understand the significance of figure 5 and the analysis conducted with the centroids. What is the meaning? what are the reasons for the displacement? does it have commercial repercussions? or is an issue related to MSP? Governance? Water quality? Production? Business development? Please try to better correlate this result with the discussion.

Response 17: Thank you very much for your very helpful comments on my manuscript. We analyzed the displacement of centroids of aquaculture ponds from 1984 to 2022 in detail. The direction of the displacement of aquaculture ponds’ centroids indicates that aquaculture will increase in that direction. Consequently, environmental factors such as water quality in this direction should be given priority to be monitored. Now we correlate this result with the discussion.

Round 2

Reviewer 4 Report

Comments and Suggestions for Authors

Dear authors

I have read your corrections and the letter but, as I understood, you have only considered some issues... 

For my point of view your discussion not clearly based on your data, you didn't discuss your results with other international studies in order to clearly highlight the relevance of your results and research.

You still leave new results in the discussion... For my point of view, new results cannot be describe in the discussion.... 

Best regards

Comments on the Quality of English Language

None

Author Response

Response to Reviewer 4 Comments

1. Summary

Thank you very much for taking the time to review this manuscript again. Your suggestions will be significantly enhanced of my work and helped me avoid some of the flaws in the manuscript. We have improved the manuscript exactly as your suggestion. Please find the detailed responses below and the corresponding revisions in the re-submitted files. Once again, thank you for your recognition and dedication to our work.

2. Questions for General Evaluation

Reviewer’s Evaluation

Response and Revisions

Does the introduction provide sufficient background and include all relevant references?

Yes/Can be improved/Must be improved/Not applicable

Are all the cited references relevant to the research?

Yes/Can be improved/Must be improved/Not applicable

Is the research design appropriate?

Yes/Can be improved/Must be improved/Not applicable

Are the methods adequately described?

Yes/Can be improved/Must be improved/Not applicable

Are the results clearly presented?

Yes/Can be improved/Must be improved/Not applicable

Are the conclusions supported by the results?

Yes/Can be improved/Must be improved/Not applicable

3. Point-by-point response to Comments and Suggestions for Authors

1.     Comments 1: I have read your corrections and the letter but, as I understood, you have only considered some issues...

Response 1: Thank you for once again reviewing our manuscript. Our team has carefully considered your suggestions and carefully improved the manuscript.

2.     Comments 2: For my point of view your discussion not clearly based on your data, you didn't discuss your results with other international studies in order to clearly highlight the relevance of your results and research.

Response 2: Thank you very much for your very helpful suggestion on my manuscript. We added some discussion about our results with other international studies in Lines 302-305; Lines 333-337, and mark in red. Some reference also added in [41] and [44]. However, studies of the relationship between coastal pond changes and coastal zone vegetation using our method remain inadequate. We have made changes to the best of our ability. We hope that you will satisfactory the content of this revision.

3.     Comments 3: You still leave new results in the discussion... For my point of view, new results cannot be describe in the discussion....

Response 3: Thank you very much for your very helpful suggestions on my manuscript. I am very sorry and this question was an oversight on my part. We have placed the new result in the discussion in section 3.5 in Lines 267-272, and marked in red. Figures 6 and 7 were also switched in the manuscript.

Your Sincerely,

Kai Liu

Reviewer 5 Report

Comments and Suggestions for Authors

The manuscript is greatly improved.

I am sorry you did not take the opportunity to add a diagram/sketch to better explain the relationship between the increase in aquaculture facilities and the advancement of the coastline.

I remain convinced that it is better to reverse the order of figures 2 and 3 .

In figure 3 it is better to write the year above each box and not just in the caption.

On line 190 I think there is a mistake. If I understand correctly the time series used goes up to the year 2022. If it is correct, the image in figure 3(i) should be of 2022 and not 2033. In this case the number 2033 in caption of Fig 3, should be changed.

International journal should have all tables with units and not just in the caption. Therefore, I invite authors to add the unit (km) not only in table 4 , but also in 1, 2 and 3 and in all parts of the manuscript where is better to produce tables independent and/or self-consistent from captions and text of the ms.

Author Response

Response to Reviewer 5 Comments

1. Summary

Thank you very much for taking the time to review this manuscript again. Your reminders will be significantly enhanced of my work and also helped me avoid some errors in the manuscript. We have improved the manuscript exactly as your reminders. Please find the detailed responses below and the corresponding revisions in the re-submitted files. Once again, thank you for your recognition and dedication to our work.

2. Questions for General Evaluation

Reviewer’s Evaluation

Response and Revisions

Does the introduction provide sufficient background and include all relevant references?

Yes/Can be improved/Must be improved/Not applicable

Are all the cited references relevant to the research?

Yes/Can be improved/Must be improved/Not applicable

Is the research design appropriate?

Yes/Can be improved/Must be improved/Not applicable

Are the methods adequately described?

Yes/Can be improved/Must be improved/Not applicable

Are the results clearly presented?

Yes/Can be improved/Must be improved/Not applicable

Are the conclusions supported by the results?

Yes/Can be improved/Must be improved/Not applicable

3. Point-by-point response to Comments and Suggestions for Authors

1.     Comments 1: I am sorry you did not take the opportunity to add a diagram/sketch to better explain the relationship between the increase in aquaculture facilities and the advancement of the coastline.

Response 1: Thank you very much for your very helpful reminder on my manuscript. We added Figure. 8 to explain the relationship between the increase in aquaculture facilities and the advancement of the coastline in Lines 318-319 and mark in red.

2.     Comments 2: I remain convinced that it is better to reverse the order of figures 2 and 3.

Response 2: Thank you very much for your very helpful reminder on my manuscript. It was an oversight on my part and I'm very sorry. We have reverse the order of Figures 2 and 3.

3.     Comments 3: In figure 3 it is better to write the year above each box and not just in the caption.

Response 3: Thank you very much for your very helpful reminder on my manuscript. It was an oversight on my part and I'm very sorry. We have added the year in the Figure. 3 (Figure. 2 in the revised manuscript).

4.     Comments 4: On line 190 I think there is a mistake. If I understand correctly the time series used goes up to the year 2022. If it is correct, the image in figure 3(i) should be of 2022 and not 2033. In this case the number 2033 in caption of Fig 3, should be changed.

Response 4: Thank you very much for your very helpful reminder. I neglected to make this little mistake, thanks for the heads up. We have modified 2033 to 2022 in the caption of Figure. 3 (Figure. 2 in the revised manuscript).

5.     Comments 5: International journal should have all tables with units and not just in the caption. Therefore, I invite authors to add the unit (km) not only in table 4, but also in 1, 2 and 3 and in all parts of the manuscript where is better to produce tables independent and/or self-consistent from captions and text of the ms.

Response 5: Thank you very much for your very helpful suggestion on my manuscript. We have added the unit (km) and (m/y) in Table 1 to Table 4 and marked in red.

Your Sincerely,

Kai Liu
